# Machine learning for predicting Chagas disease infection in rural areas of Brazil

**Fabio De Rose Ghilardi** [1] *, **Gabriel Silva**[2], **Thallyta Maria Vieira**[3], **Ariela Mota**[3], **Ana Luiza Bierrenbach**[1], **Renata Fiuza Damasceno**[3], **Lea Campos de Oliveira**[4], **Alexandre Dias Porto Chiavegatto Filho**[2], **Ester Sabino**[1,4]

1 Faculdade de Medicina da Universidade de São Paulo–FMUSP, São Paulo, Brazil, 2 Faculdade de Saúde Pública da Universidade de São Paulo–FSP USP, São Paulo, Brazil, 3 Universidade Estadual de Montes Claros–Unimontes, Montes Claros, Minas Gerais, Brazil, 4 Instituto de Medicina Tropical da Faculdade de Medicina da USP–IMT USP, São Paulo, Brazil

* fabio.ghilardi@hc.fm.usp.br

**Data Availability Statement:** The code developed for constructing the algorithms along with the original dataset, is available on Github (https://github.com/fabioghilardi/chagas_disease_

## Abstract

### Introduction

Chagas disease is a severe parasitic illness that is prevalent in Latin America and often goes unaddressed. Early detection and treatment are critical in preventing the progression of the illness and its associated life-threatening complications. In recent years, machine learning algorithms have emerged as powerful tools for disease prediction and diagnosis.

### Methods

In this study, we developed machine learning algorithms to predict the risk of Chagas disease based on five general factors: age, gender, history of living in a mud or wooden house, history of being bitten by a triatomine bug, and family history of Chagas disease. We analyzed data from the Retrovirus Epidemiology Donor Study (REDS) to train five popular machine learning algorithms. The sample comprised 2,006 patients, divided into 75% for training and 25% for testing algorithm performance. We evaluated the model performance using precision, recall, and AUC-ROC metrics.

### Results

The Adaboost algorithm yielded an AUC-ROC of 0.772, a precision of 0.199, and a recall of 0.612. We simulated the decision boundary using various thresholds and observed that in this dataset a threshold of 0.45 resulted in a 100% recall. This finding suggests that employing such a threshold could potentially save 22.5% of the cost associated with mass testing of Chagas disease.

### Conclusion

Our findings highlight the potential of applying machine learning to improve the sensitivity and effectiveness of Chagas disease diagnosis and prevention. Furthermore, we emphasize the importance of integrating socio-demographic and environmental factors into neglected disease prediction models to enhance their performance.

screening; https://github.com/gabriel1710/chagas_disease_prediction).”

**Funding:** The author(s) received no specific funding for this work.

**Competing interests:** The authors have declared that no competing interests exist.

## Author summary

Chagas disease, a severe parasitic illness prevalent in Latin America, poses significant challenges due to delayed detection and treatment. Machine learning algorithms, advanced computer programs, have emerged as valuable tools for disease prediction and diagnosis. In our study, we utilized these algorithms to forecast Chagas disease risk based on factors such as age, gender, and living conditions. Drawing on data from the Retrovirus Epidemiology Donor Study (REDS), we trained five algorithms, with one showing promising results, achieving an impressive score of 0.772 out of 1. By establishing a specific threshold, we could potentially reduce testing costs while maintaining high detection rates. This research highlights the potential of machine learning in improving Chagas disease diagnosis and prevention by incorporating socio-demographic and environmental factors. Integrating these elements into predictive models has the potential to enhance their effectiveness and sensitivity, thereby improving disease management outcomes and ultimately reducing the burden of Chagas disease in affected regions.

## Introduction

Chagas disease (CD), also known as American trypanosomiasis, is a parasitic disease caused by the protozoan parasite Trypanosoma cruzi. It is prevalent in Latin America, where an estimated 6–7 million people currently infected [1]. CD is transmitted primarily through infected triatomine bugs, which are commonly known as "kissing bugs" due to their tendency to bite around the lips and face. However, the disease can also be transmitted through blood transfusions, organ transplants, and from mother to child during pregnancy [2].

Despite being a neglected tropical disease, CD has gained increasing attention in recent years due to its potential to spread beyond Latin America. With globalization and increased travel, CD has also been reported in non-endemic areas, including the United States, Canada, Europe, and Japan [3]. Furthermore, the chronic nature of the disease and the lack of effective treatments have resulted in significant economic and health burdens for affected individuals and communities [4].

Efforts to control and prevent CD have primarily centered around vector control, blood screening, and education [5]. However, resource challenges persist, particularly in remote regions of Latin America where access to diagnosis and treatment is limited. There is a pressing need to incorporate more effective screening strategies into Public Health Care Facilities. To address this, we conducted an evaluation of an artificial intelligence- based screening tool that utilizes simple and easily applicable questions. Our objective was to improve the identification of individuals with CD, particularly in resource-limited areas.

## Methods

### Ethics statement

This study was conducted in accordance with the guidelines of the Institutional Review Board of Universidade Estadual de Montes Claros. Approval was granted under protocol number CAAE 02279618.0.0000.5146. Formal written consent was obtained from all participants before their involvement in the study.

## Questionnaire development

A detailed health history and a socio demographic questionnaire came from a data set of a previous study [6] (REDS Retrovirus Epidemiology Donor Study, Sabino et al; 2013). This database includes first time blood donors constituted by 500 patients with a positive serology for CD, 500 patients with a negative serology for CD (our control sample) and 100 patients with chagas cardiomyopathy.

The dataset was collected from visits by community health agents to the municipalities of Espinosa and São Francisco, both located in the state of Minas Gerais, Brazil. During these visits, questions were asked about the patient's age, gender, family history of CD, and housing history.

The initial dataset included 2,061 patients, of which 196 (9.5%) had a positive outcome for CD, 1,810 (87.8%) had a negative outcome, and 55 had an inconclusive result (2.7%). The ELISA serology test, which is a laboratory technique that uses antibodies and enzymes to detect the presence of a specific antigen in a biological sample, was used for diagnosis. It is widely used in clinical and research laboratories to detect a large variety of antigens, such as proteins, hormones, viruses, and bacteria. In CD, the ELISA serology test is used to detect the presence of antibodies against *Trypanosoma cruzi* in patient blood.

## Outcome definition

The outcome was obtained from the ELISA serology test result. Variables were collected to analyze if the patient presents a risk of having CD. Thus, the predicted outcome was the risk of the patient having CD, further requiring referral for confirmation of the diagnostic hypothesis through serology.

## Data preprocessing and machine learning algorithms

Patients with inconclusive results (55) were removed from the sample. Inconclusive patients refer to ELISA results that were beyond the diagnosis threshold used in the KIT test. In certain situations, the decrease in values of serological markers can be attributed to a possible resolution of the infection, or it could be a result of cross-reaction with serology related to other prevalent infections in the area. For instance, patients with a history of previous leishmaniasis infection, which is common in the northern region of Minas Gerais where the study was conducted, may exhibit cross-reactivity. To mitigate the potential bias associated with an incorrect diagnosis of CD, we excluded these inconclusive serology results from our analysis. After the removal of inconclusive serologies, the dataset comprised a total of 2,006 patients.

Categorical variables (family history of CD, relationship of family member with CD, previous residence in a region with kissing bugs, living in a wooden or mud house, and being bitten by a kissing bug) were subjected to one-hot encoding, while the only quantitative variable (age) was normalized via z-score. For the treatment of missing values, mean imputation was applied to the training set. Categorical variables were treated via the most frequent value (relationship of family member with CD) imputation or via categorical missing (residence in a region with kissing bugs, living in a wooden or mud house, and being bitten by a kissing bug), where missing values were considered an additional category after applying one-hot encoding.

Due to the imbalance of the dataset (9.8% of patients had CD after removing inconclusive results), the Synthetic Minority Oversampling TEchnique (SMOTE) was applied to the training data [7] (Chawla et al., 2002). Five popular machine learning algorithms were tested: Adaboost [8](Schapire, 2013), LightGBM [9] (Ke et al., 2017), XGBoost [10] (Chen & Guestrin, 2016), Catboost [11] (Prokhorenkova et al., 2018), and Random Forest [12] (Ho, 1995).

The training strategy used was a hold-out with 75% for training and 25% for testing. Hyperparameters were optimized by the RandomSearch technique [13] (Bergstra & Bengio, 2012), considering stratified 10-fold cross-validation in the 75% training set. The evaluation of algorithm performance was performed exclusively on the test set, based on the following metrics: area under the ROC curve (AUC-ROC), precision, recall, F1-score, and accuracy. For the interpretation of the decision-making process of the algorithms, the Shapley values technique was used for LightGBM, XGBoost, Catboost, and Random Forest. Since the Adaboost algorithm does not allow for the implementation of Shapley values, the technique of variable importance by random permutation was used in this case.

We conducted a cost simulation from different algorithmic decision thresholds. Therefore, the thresholds 0.20, 0.25, 0.30, 0.35, 0.40, 0.45, 0.50, 0.55, 0.60, 0.65, 0.60, 0.65, and 0.70 were tested. The objective of this simulation was to evaluate the possibility of finding a threshold where the algorithm's sensitivity was 100% and then to estimate what the savings of public resources would be compared to testing all patients. In this case, the test data (502 patients) was used in the algorithm with the highest AUC-ROC.

We followed the guidelines of the Transparent Reporting of a multivariable prediction model for Individual Prognosis or Diagnosis (TRIPOD) [14] (Moons et al., 2015). The code developed for constructing the algorithms along with the original dataset, is available on Github (https://github.com/fabioghilardi/chagas_disease_screening; https://github.com/gabriel1710/chagas_disease_prediction). The collection of primary data obtained during the process was approved by the Ethics Committee of the State University of Montes Claros–UNIMONTES (CAAE: 02279618.0.00005146).

## Results

### Descriptive analysis

After data preprocessing, a total of 2,006 patients were analyzed. About 36.6% of the sample consisted of men and 63.4% of women. A total of 43.4% of patients had a family history of CD, 28.3% lived in a house made of wood or mud, and 15.1% lived in a region with the presence of triatomine bugs. Regarding triatomine bug bites, 25.1% of patients reported being bitten and 38.9% did not report being bitten. Notably, a large proportion of patients (35.8%) did not know if they had been bitten or not. Overall, the proportions in the training and testing sets remained close to those observed in the complete dataset (Table 1).

### Algorithms performance

The Adaboost algorithm with its default hyperparameter configuration presented a 0.772 AUC-ROC (Table 2). In general, all the algorithms presented at least one combination of techniques with an AUC-ROC above 0.700. It is important to clarify that we are evaluating different algorithms in terms of their performance. Accuracy has important limitations in unbalanced scenarios but quantifies the percentage of correct classifications attained by a trained machine learning algorithm. Precision, on the other hand, serves as an indicator of the algorithm ability to accurately make positive predictions. It is computed by dividing the number of true positives by the total number of positive predictions. Additionally, recall, which is the same as sensitivity, indicates the proportion of positive samples correctly classified as positive relative to the total number of positive samples.

For Adaboost, the algorithm with the best performance regarding AUC-ROC and sensitivity, we plotted the ROC curves for the three different technique combinations (Fig 1). In general, the behavior of the three curves was similar, reflecting their overall similar AUC-ROC

**Table 1. Descriptive Summary of Full, Train, and Test Datasets Used for Developing Algorithms in Chagas Disease Classification.**

| Variable | Full Dataset | Train (75%) | Test (25%) |
|---|---|---|---|
| Sample Size | 2,006 | 1,504 | 502 |
| Outcome | | | |
| Positive | 196 (9.8%) | 147 (9.8%) | 49 (9.8%) |
| Negative | 1,810 (90.2%) | 1,357 (90.2%) | 453 (90.2%) |
| Sex | | | |
| Male | 735 (36.6%) | 560 (37.2%) | 175 (34.9%) |
| Female | 1,271 (63.4%) | 944 (62.8%) | 327 (65.1%) |
| Family History of CD | | | |
| Yes | 870 (43.4%) | 655 (43.6%) | 215 (42.8%) |
| No | 1,136 (56.6%) | 849 (56.4%) | 287 (57.2%) |
| Brother or sister | 27 (1.3%) | 18 (1.2%) | 9 (1.8%) |
| Grandmother/grandfather | 93 (4.6%) | 71 (4.7%) | 22 (4.4%) |
| Aunt or uncle | 303 (15.1%) | 229 (15.2%) | 74 (14.7%) |
| Father | 457 (22.8%) | 342 (22.7%) | 115 (22.9%) |
| Mother | 505 (25.2%) | 381 (25.3%) | 124 (24.7%) |
| Wooden house or wattle and daub | | | |
| Yes | 568 (28.3%) | 434 (28.9%) | 134 (26.7%) |
| No | 1,433 (71.4%) | 1,066 (70.9%) | 367 (73.1%) |
| Unassigned | 5 (0.0%) | 4 (0.0%) | 1 (0.0%) |
| Lived in a region with kissing bug | | | |
| Yes | 303 (15.1%) | 231 (15.4%) | 72 (14.3%) |
| No | 1,655 (82.5%) | 1,236 (82.2%) | 419 (83.5%) |
| Do not know | 47 (2.3%) | 36 (2.4%) | 11 (2.2%) |
| Unassigned | 1 (0.0%) | 1 (0.0%) | 0 (0.0%) |
| Stung by the kissing bug | | | |
| Yes | 503 (25.1%) | 368 (24.5%) | 135 (26.9%) |
| No | 781 (38.9%) | 589 (39.2%) | 192 (38.2%) |
| Do not know | 718 (35.8%) | 545 (36.2%) | 173 (34.5%) |
| Unassigned | 4 (0.2%) | 2 (0.1%) | 2 (0.4%) |

values (between 0.752 and 0.772). This behavior reinforces the need to make the decision regarding the best algorithm considering the other performance metrics, such as sensitivity.

## Decision interpretation and feature importance

Regarding the interpretation of algorithmic decision-making, the importance of variables via random permutation indicated that living in a wooden or mud house, along with age and being bitten by the triatomine bug, were the most important factors in the decision-making process of Adaboost (Fig 2). Factors such as paternal history of CD and patients being unsure if they were bitten or not by the bug were less significant.

In addition to the absolute average values of variable importance for Adaboost, we also developed the shapley values plot for the XGBoost algorithm with SMOTE (Fig 3). It is possible to observe that the age variable was the most important in the decision- making process of XGBoost, with higher values of this variable tending to imply higher prediction values. The same behavior can be observed for variables that refer to the family history of CD (brother or sister) and when patients do not know if they lived in a region with a kissing bug.

**Table 2. Predictive Performance of Trained Algorithms: (1) Default Version, (2) Hyperparameter Optimization, and (3) Random Search plus Hyperparameter Optimization, in Chagas Disease Classification: Results on the Test Set.**

| Model | Accuracy | Precision | Recall | AUC(ROC) | F1-Score |
|---|---|---|---|---|---|
| Adaboost default | 0.721 | 0.199 | 0.612 | **0.772** | 0.300 |
| XGBoost + SMOTE + Tuning | 0.560 | 0.169 | 0.898 | 0.771 | 0.285 |
| Catboost default | 0.900 | 0.429 | 0.061 | 0.768 | 0.107 |
| Catboost + SMOTE + Tuning | 0.823 | 0.244 | 0.388 | 0.757 | 0.299 |
| Adaboost + Tuning | 0.916 | 1.000 | 0.143 | 0.757 | 0.250 |
| LightGBM + Tuning | 0.875 | 0.294 | 0.204 | 0.755 | 0.241 |
| XGBoost default | 0.898 | 0.444 | 0.163 | 0.753 | 0.239 |
| Random Forest + SMOTE + Tuning | 0.725 | 0.165 | 0.449 | 0.752 | 0.242 |
| Adaboost + SMOTE + Tuning | 0.502 | 0.155 | **0.918** | 0.752 | 0.265 |
| LightGBM + Tuning | 0.896 | 0.421 | 0.163 | 0.748 | 0.235 |
| LightGBM + SMOTE + Tuning | 0.576 | 0.167 | 0.837 | 0.745 | 0.278 |
| XGBoost + Tuning | 0.902 | 0.000 | 0.000 | 0.725 | 0.000 |
| Catboost + Tuning | 0.902 | 0.000 | 0.000 | 0.721 | 0.000 |
| Random Forest default | 0.853 | 0.162 | 0.122 | 0.662 | 0.140 |
| Random Forest + Tuning | 0.851 | 0.158 | 0.122 | 0.574 | 0.138 |

In Fig 4, the decision paths are presented for four patients based on their Shapley values: a true positive (a), a true negative (b), a false positive (c), and a false negative (d). It is possible to observe that the effect of the numerical variable (age) is highly relevant in all four scenarios. For the true positive patient, answering "don't know" to the question about family history

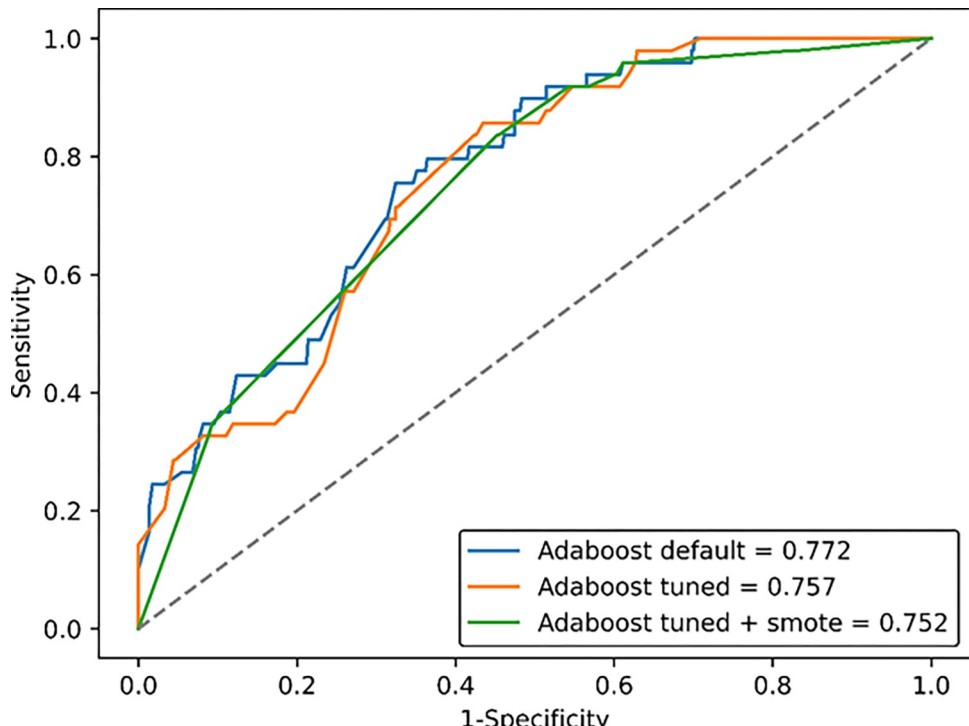

**Fig 1. Area under the ROC Curve for Adaboost Algorithm's Three Training Strategies in Chagas Disease Classification: Results on the Testing Dataset.**

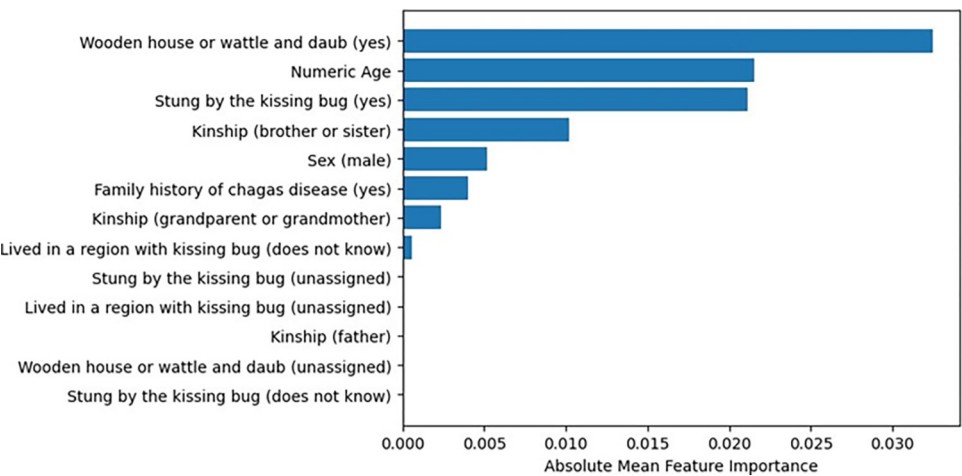

**Fig 2. Feature Importance by Random Permutation for Adaboost Default Algorithm in Chagas Disease Classification: Results on the Testing Dataset.**

increased the prediction value for the positive outcome. In the true negative patient, the fact that the patient did not live in a wooden or wattle and daub house and was not bitten by the triatomine bug were factors that decreased the prediction value. In the false positive patient, it was observed that the fact that the patient was bitten by the triatomine bug and lived in a wooden or wattle and daub house led the algorithm to classify them as having a positive outcome, although the ELISA test result was negative. Finally, in the false negative patient, although the patient did not live in a wooden or wattle and daub house and had an age that decreased their prediction value, the ELISA test result was positive.

## Avoided cost for different thresholds

We simulated the algorithmic behavior for different threshold values to identify if in any scenario all patients with CD would be classified with a positive outcome. The objective of this analysis was to understand how much could be saved with the use of the algorithm, assuming that all patients are referred for ELISA serology testing, while maintaining a sensitivity of 100% in the test set. In Fig 5, it is possible to observe that for a threshold of 0.45, the default Adaboost algorithm presents a sensitivity of 100% with a precision of 0.126.

In this scenario, 389 patients would be referred to ELISA serology testing, of which 340 would have a false negative result. However, compared to the hypothesis that all 502 patients in the test set are referred for ELISA serology testing, the cost based on the algorithmic recommendation would be approximately 22.5% lower than the global cost (all tested patients), while maintaining sensitivity at 100%. For the default threshold (0.500), referring patients with a risk above the probabilistic limit to serology testing would imply an avoided cost of approximately 70%. However, the sensitivity would be reduced to 61.22%, meaning that the efficiency of identifying patients who truly require the test would be lower.

## Discussion

The present study developed machine learning algorithms to predict the risk of a patient having Chagas disease by considering five epidemiological aspects: the patient's family history of CD, their relation to an affected family member, living in an area infested by triatome bugs, living in a house made of wooden or wattle or daub, and being bitten by a kissing bug. These

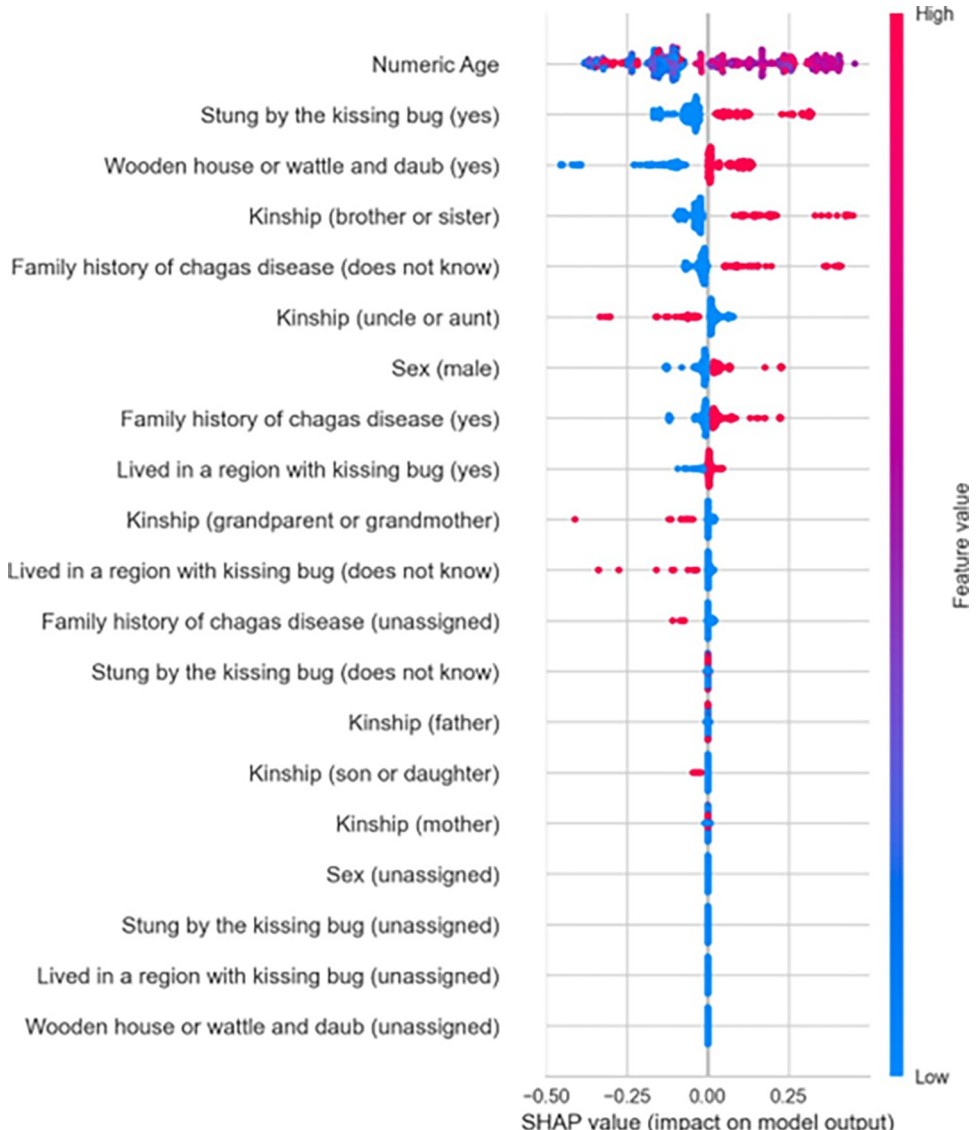

**Fig 3. Beeswarm shapley-values plot for the XGBoost tuned with SMOTE in Chagas Disease Classification: Results on the Testing Dataset.**

findings are supported by a limited number of previous studies and are in line with the American recommendations for CD screening and diagnosis in the United States. These recommendations were developed by a diverse group of clinicians and researchers with expertise in CD. Our study provides substantial evidence that supports the decision to include these risk factors in the screening in diagnosis protocol.

The results obtained in this study provide positive evidence for the use of machine learning in predicting CD diagnosis. Considering that the data comes from questionnaires collected from visits by community health agents and from five general questions, the predictive performances of 0.772 for AUC-ROC and 0.918 for sensitivity highlight the future use of these algorithms as a promising tool in identifying patients with CD. However, it is important to note that these results are still preliminary and need to be validated in future studies with a larger sample.

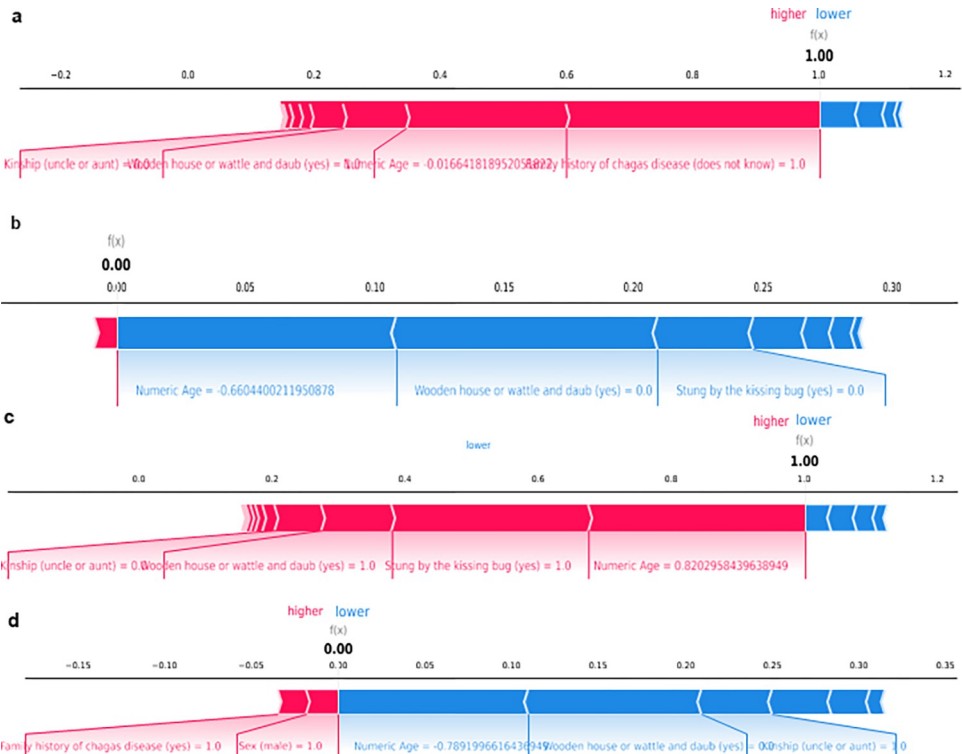

**Fig 4.** Shapley-Values Force-Plot for Different Patient Types in Chagas Disease Classification Using Adaboost Default Algorithm: (a) True-Positive Patient, (b) True- Negative Patient, (c) False-Positive Patient, and (d) False-Negative Patient. Results Based on Testing Dataset.

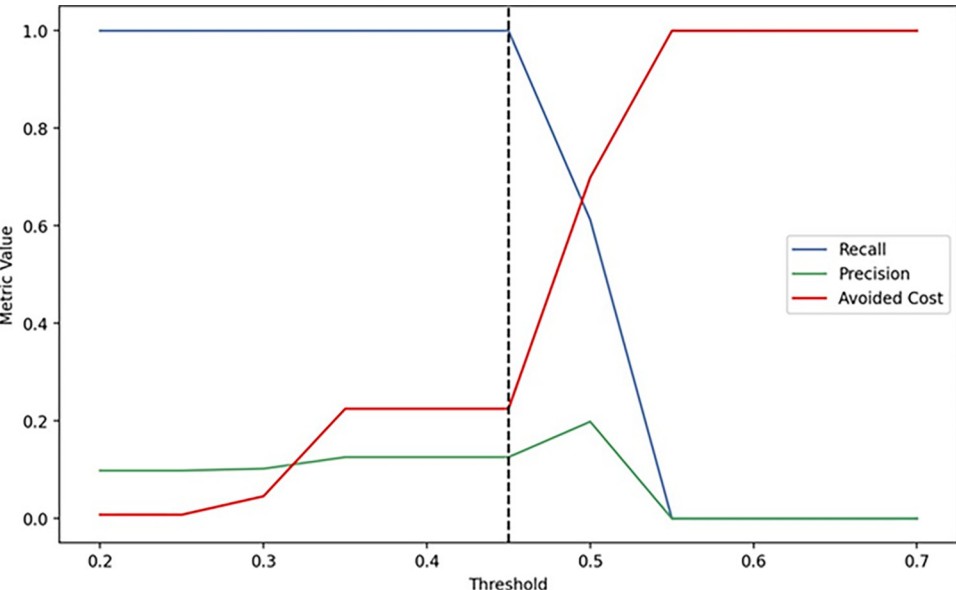

**Fig 5. Simulations of Precision, Recall, and Cost Avoidance with Varying Threshold Values for Adaboost Default Algorithm in Chagas Disease Classification: Results from the Testing Dataset.**

The preference for sensitivity (recall) is an important choice, as it can help identify all positive cases of CD, even in the face of negative cases incorrectly classified as positive. Changes in the decision threshold allowed the identification of scenarios with 100% sensitivity, an important preliminary discovery for this population. In addition to the possibility of identifying all positive cases of CD, even in the face of false positives, it is possible to use the algorithm as a screening and referral tool for ELISA serology, representing a saving of public resources of around 22.5%.

In a future potential practical application, considering that the data was collected during periodic visits made by community health agents, the algorithm could be used during this same visit after data collection and completion. In the absence of technology at the time of the visit, such as a cell phone or tablet, which is common in Brazil, the community agents could collect the data and the municipal health surveillance team later apply the predictive algorithm.

It is worth noting that all the risk factors discussed are related to vectorial transmission in rural areas, where lack of resources and poor housing conditions increase the risk of human contact with the bugs. However, the same questionnaire may not be effective for other transmission routes, such as oral transmission, which is often associated with outbreaks caused by contaminated beverages. Although CD transmission through blood transfusions is infrequent, the present questionnaire questions could also be improved to evaluate blood donors effectively. Moreover, the questionnaire can be an essential tool in identifying pregnant women with CD to avoid mother-to-child transmission, especially in areas with low disease prevalence.

Despite the promising results, this work presents important limitations. The first limitation is the small sample size and the specificity of the results for the population in question. Although this is a point that does not invalidate the results, the generalization of its performance in new data and other populations should be carefully evaluated. Another concern in regarding the inconclusive tests, which represented approximately 2.7% of the initial sample. Although it is a relatively low value, the exclusion of these patients from the predictive model may represent biases in future implementations of the algorithm.

It is also important to recognize that the same questions may not perform uniformly across different regions, even in Latin America. Environmental conditions and the presence of other types of vectors can have an impact on vectorial capacity. For example, different types of triatomideae may be more effective at transmitting the disease, and the density of other hosts could also impact disease spread. Therefore, future studies must consider these factors to develop more effective prevention and control measures.

The results found by this study represent preliminary scientific findings that demonstrate the feasibility of the use of machine learning algorithms as a useful tool in diagnosing patients with CD and prioritizing referrals for the ELISA serology based solely on questionnaire data.

## Supporting information

**S1 Data. Chagas Screening Data.** This CSV file contains raw screening data for Chagas disease, which were collected and analyzed in the study. It includes test results, demographic information, and other relevant variables used in the data analysis. This supplementary material is provided to ensure transparency and to allow for replication of the study's findings by independent researchers.
(CSV)

## Author Contributions

**Conceptualization:** Alexandre Dias Porto Chiavegatto Filho, Ester Sabino.

**Data curation:** Fabio De Rose Ghilardi, Gabriel Silva, Thallyta Maria Vieira, Ariela Mota, Ana Luiza Bierrenbach, Renata Fiuza Damasceno, Lea Campos de Oliveira.

**Formal analysis:** Fabio De Rose Ghilardi, Gabriel Silva.

**Funding acquisition:** Alexandre Dias Porto Chiavegatto Filho, Ester Sabino.

**Investigation:** Fabio De Rose Ghilardi, Gabriel Silva.

**Methodology:** Fabio De Rose Ghilardi, Gabriel Silva.

**Project administration:** Fabio De Rose Ghilardi.

**Resources:** Alexandre Dias Porto Chiavegatto Filho, Ester Sabino.

**Software:** Fabio De Rose Ghilardi, Gabriel Silva.

**Supervision:** Alexandre Dias Porto Chiavegatto Filho, Ester Sabino.

**Validation:** Fabio De Rose Ghilardi, Gabriel Silva.

**Visualization:** Fabio De Rose Ghilardi, Thallyta Maria Vieira, Ariela Mota, Ana Luiza Bierrenbach, Renata Fiuza Damasceno, Lea Campos de Oliveira.

**Writing – original draft:** Fabio De Rose Ghilardi, Gabriel Silva.

**Writing – review & editing:** Fabio De Rose Ghilardi.

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
