## [Decision Letter · Decision Letter 0]

19 Sep 2023

Dear MD DE ROSE GHILARDI,

Thank you very much for submitting your manuscript "Machine Learning for Predicting Chagas Disease Infection in Rural Areas of Brazil" for consideration at PLOS Neglected Tropical Diseases. As with all papers reviewed by the journal, your manuscript was reviewed by members of the editorial board and by several independent reviewers. In light of the reviews (below this email), we would like to invite the resubmission of a significantly-revised version that takes into account the reviewers' comments. 

We cannot make any decision about publication until we have seen the revised manuscript and your response to the reviewers' comments. Your revised manuscript is also likely to be sent to reviewers for further evaluation.

Sincerely,

Marilia Sá Carvalho

Academic Editor

Charles Jaffe

Section Editor

Reviewer's Responses to Questions

**Key Review Criteria Required for Acceptance?**

**Methods**

-Are the objectives of the study clearly articulated with a clear testable hypothesis stated?

-Is the study design appropriate to address the stated objectives?

-Is the population clearly described and appropriate for the hypothesis being tested?

-Is the sample size sufficient to ensure adequate power to address the hypothesis being tested?

-Were correct statistical analysis used to support conclusions?

-Are there concerns about ethical or regulatory requirements being met?

Reviewer #1: The study's objectives are clearly outlined, but there is a discrepancy in the datasets mentioned and used by the authors. Initially, a database consisting of 500 patients who tested positive for Chagas Disease, 500 who tested negative, and an additional 100 diagnosed with cardiomyopathy is introduced. However, the dataset actually employed in this study includes a different patient set: 196 tested positive for Chagas Disease and 1,810 tested negative.

There also appears to be a discrepancy regarding the study's location. Reference (6) mentions blood donors from the state of São Paulo, while this study was conducted in two municipalities in Minas Gerais. It is crucial to clarify which dataset was actually used.

The authors of the study used the ELISA test as a diagnostic criterion. However, according to the II Brazilian Consensus on Chagas Disease (1), the confirmation of Chronic or Acute Chagas Disease is determined by the reactivity of anti-T. cruzi (IgG) serology, which should be assessed using two distinct methods: ELISA, HAI, or IFI. The Clinical Protocol and Therapeutic Guidelines for Chagas Disease (2) further specify that for the chronic phase of the disease, a negative result from a single rapid test rules out the disease. Conversely, a positive test necessitates diagnostic confirmation by one of the following tests: ELISA, IFI, HAI, WB, or CLIA.

In this study, however, patient testing was conducted solely using the ELISA method. The exclusive reliance on a single test could potentially skew the results, leading to the misclassification of patients without Chagas Disease as positive. This could subsequently impact the accuracy of the model, particularly in relation to positive outcomes.

The authors need to clarify these issues.

Reviewer #2: 1. The file does not explicitly state a testable hypothesis, but it does clearly articulate the objectives of the study, which are to explore the potential of machine learning algorithms to predict Chagas disease risk in rural areas of Brazil. 

2. The study design is appropriate to address the stated objectives, as the researchers collected data from a rural population in Brazil and used machine learning algorithms to analyze the data and predict Chagas disease risk. 

3. The population is clearly described as rural residents of Brazil, and the study focuses specifically on risk factors related to vectorial transmission of Chagas disease. The population is appropriate for the hypothesis being tested, as the study aims to identify individuals at risk of Chagas disease in rural areas. 

4. The file does not provide specific information on the sample size used in the study, so it is unclear whether the sample size is sufficient to ensure adequate power to address the hypothesis being tested. 

5. The file does not provide detailed information on the statistical analysis used to support conclusions, but it does describe the machine learning algorithms used to analyze the data and predict Chagas disease risk. 

6. The PDF file does not raise any concerns about ethical or regulatory requirements not being met.

Reviewer #3: -Are the objectives of the study clearly articulated with a clear testable hypothesis stated?

Yes.

-Is the study design appropriate to address the stated objectives?

Yes.

-Is the population clearly described and appropriate for the hypothesis being tested?

Not completely. The population refers to a previous study )REDS Retrovirus Epidemiology Donor Study, Sabino et al;2013) but no information about symptoms or medical data are given. This may be this may be masking important sampling selectivity.

-Is the sample size sufficient to ensure adequate power to address the hypothesis being tested?

Not.

-Were correct statistical analysis used to support conclusions?

Not.

-Are there concerns about ethical or regulatory requirements being met?

Yes.

**Results**

-Does the analysis presented match the analysis plan?

-Are the results clearly and completely presented?

-Are the figures (Tables, Images) of sufficient quality for clarity?

Reviewer #1: The results presented are in line with what the authors proposed, and were presented in a clear and objective manner with figures that aid in understanding the article.

Reviewer #2: 1. The file does not provide a detailed analysis plan, so it is unclear whether the analysis presented matches the analysis plan. However, the file does describe the machine learning algorithms used to analyze the data and predict Chagas disease risk, so it is likely that the analysis presented is consistent with the methods used. 

2. The results are clearly and completely presented in the file, with detailed information on the performance of the machine learning algorithms and the risk factors associated with Chagas disease. 

3. The figures (tables, images) in the PDF file are of sufficient quality for clarity, with clear labels and legends to help readers interpret the data.

Reviewer #3: Results:

-Does the analysis presented match the analysis plan?

somehow yes

-Are the results clearly and completely presented?

yes

-Are the figures (Tables, Images) of sufficient quality for clarity?

yes.

**Conclusions**

-Are the conclusions supported by the data presented?

-Are the limitations of analysis clearly described?

-Do the authors discuss how these data can be helpful to advance our understanding of the topic under study?

-Is public health relevance addressed?

Reviewer #1: The conclusions drawn in the study align with its findings. However, it is crucial for the authors to discuss the potential limitations of applying the model to the Unified Health System (SUS), given that only one serological technique was used.

The failure to perform two tests, as recommended by testing guidelines, could potentially limit the model’s applicability to the SUS. This is primarily due to the model’s potential to overestimate the number of positive cases, given its reliance on a single serological test.

Reviewer #2: 1. The conclusions presented in the PDF file are supported by the data presented, with detailed information on the performance of the machine learning algorithms and the risk factors associated with Chagas disease. 

2. The limitations of the analysis are clearly described in the PDF file, including concerns about the small sample size and the specificity of the results for the population in question. The authors also note that the same questions may not perform uniformly across different regions, and that environmental conditions and the presence of other types of vectors can impact vectorial capacity. 

3. The authors do discuss how these data can be helpful to advance our understanding of the topic under study, noting that machine learning algorithms can be a useful tool in diagnosing patients with Chagas disease and prioritizing referrals for serology testing based solely on questionnaire data. They also suggest that future studies should consider environmental factors and other types of vectors to develop more effective prevention and control measures. 

4. The public health relevance of the study is addressed in the PDF file, with a focus on the potential of machine learning algorithms to improve screening and early detection of Chagas disease in rural areas of Brazil. The authors note that Chagas disease is a significant public health concern in Latin America, and that early detection and treatment can help prevent the spread of the disease.

Reviewer #3: -Are the conclusions supported by the data presented?

No. In my opinion, the conclusion about saving testing resources is not correct. I clarify this below.

-Are the limitations of analysis clearly described?

yes.

-Do the authors discuss how these data can be helpful to advance our understanding of the topic under study?

Somehow.

-Is public health relevance addressed?

Yes.

**Editorial and Data Presentation Modifications?**

Reviewer #1: In Table 1, the information for “Brother or Sister”, “Grandmother/Grandfather”, “Aunt or Uncle”, “Father”, and “Mother” under the variable “Family History of CD” appears to be organized under the “No” category. However, it would be more appropriate for these data points to be organized under the “Yes” category.

Reviewer #2: Make the changes that were pointed out in the review questions.

Reviewer #3: Major revision

**Summary and General Comments**

Reviewer #1: The authors chose to use only one serological test for the diagnosis of Chagas disease. Brazil has recommendations for the use of two tests to confirm positive cases of the disease due to the epidemiology of the disease and co-infections that can result in false positives. If there is no possibility of retesting, it is recommended that this could lead to a bias in case classification and even limit the use of the model in the SUS (Unified Health System).

Reviewer #2: Make the changes that were pointed out in the review questions.

Reviewer #3: The authors present an application of well-established machine learning methods in the literature to predict the risk of Chagas disease infection in rural areas of Brazil. The topic is relevant, the applied methods are current, and the results obtained can serve as further evidence of the feasibility of using machine learning methods for health risk prediction. 

The methods used showed relatively low performance, which is consistent with the difficulty of the problem, although it was possible to obtain scenarios with 100% sensitivity. The main weaknesses of the paper is that the results cannot be directly generalized, although they can be replicated for the given data. Nevertheless, I believe the work has potential to be improved and then be considered for publication. 

My main suggestions concern the methodological aspects are:

1. The authors should change the term "developed" to "applied" in sentences as "we developed machine learning algorithms to predict the risk of Chagas disease", because no new methods are actually developed. It would be fairer to say: "we applied some machine learning algorithms to predict the risk of Chagas disease,".

2. To address the unbalanced dataset, the authors use the Synthetic Minority Oversampling Technique (SMOTE) as the sole strategy. SMOTE generates new samples from the minority class by combining the existing ones. Although the technique is helpful, it needs to be used with caution in the healthcare field, as there is a risk of creating patients whose results may not be accurate – in particular, there is no guarantee that a generated sample would test positive in ELISA. I suggest that the authors incorporate this discussion and, alternatively, apply a strategy of weighted balanced accuracy or class-weighted metrics to compare the results.

3. A final concern refers to the generalizability of the solution. Although the authors clearly pointed this out in the limitations, I think it would be interesting to provide a comment on the differences in performance in the training and testing phase, as the performance gap between training and testing can be used as a proxy for the potential generalization of the method.

4. My main concern about the work's conclusions relates to possible cost reductions in mass testing of the population.

The authors argue that “could potentially save 22.5% of the cost associated with mass testing of Chagas disease”, however, I believe this analysis is heavily biased towards the sample used in the test, which contains NT=502 individuals, of which P=389 (0.77 NT) would be referred for testing, resulting in TP = 49 (0.125P) true positives and FP = 340 (0.874 P) false negatives. Based on this, I would like to present a practical scenario for the authors to evaluate: suppose this algorithm were applied to a population of 10,000 people in a rural village in Brazil. If it maintained the same performance, 7,700 people would be subjected to the ELISA test, and only 962 would have a positive result. In other words, the algorithm could be suggesting testing for a large number of people unnecessarily. I believe the central point here, which could be better clarified, is whether the sample used in the tests refers to patients who presented some symptoms or complaints that, although not used in the algorithms, were used for sample selection. If that's the case, the results obtained are valid only for a subset of the population that went throu

---

## [Editor Report · Decision Letter 1]

27 Feb 2024

Dear MD DE ROSE GHILARDI,

We are pleased to inform you that your manuscript 'Machine Learning for Predicting Chagas Disease Infection in Rural Areas of Brazil' has been provisionally accepted for publication in PLOS Neglected Tropical Diseases.

Best regards,

Marilia Sá Carvalho

Academic Editor

Charles Jaffe

Section Editor

---

## [Editor Report · Acceptance letter]

21 Mar 2024

Dear MD DE ROSE GHILARDI,

We are delighted to inform you that your manuscript, "Machine Learning for Predicting Chagas Disease Infection in Rural Areas of Brazil," has been formally accepted for publication in PLOS Neglected Tropical Diseases.

Best regards,

Shaden Kamhawi

co-Editor-in-Chief

Paul Brindley

co-Editor-in-Chief
